# Changes in the Organosulfur and Polyphenol Compound Profiles of Black and Fresh Onion during Simulated Gastrointestinal Digestion

**DOI:** 10.3390/foods10020337

**Published:** 2021-02-04

**Authors:** Alicia Moreno-Ortega, José Luis Ordóñez, Rafael Moreno-Rojas, José Manuel Moreno-Rojas, Gema Pereira-Caro

**Affiliations:** 1Department of Food Science and Health, Andalusian Institute of Agricultural and Fisheries Research and Training (IFAPA), Alameda del Obispo, Avda. Menéndez-Pidal, s/n., 14004 Córdoba, Spain; josel.ordonez@juntadeandalucia.es (J.L.O.); josem.moreno.rojas@juntadeandalucia.es (J.M.M.-R.); 2Departamento de Bromatología y Tecnología de los Alimentos, Campus Rabanales, Ed. Darwin-anexo Universidad de Córdoba, 14014 Córdoba, Spain; rafael.moreno@uco.es

**Keywords:** black onion, fresh onion, polyphenols, organosulfur compounds, simulated digestion, in-vitro digestion, bioaccessibility

## Abstract

This study aims to determine the changes in, and bioaccessibility of, polyphenols and organosulfur compounds (OSCs) during the simulated gastrointestinal digestion of black onion, a novel product derived from fresh onion by a combination of heat and humidity treatment, and to compare it with its fresh counterpart. Fresh and black onions were subjected to in-vitro gastrointestinal digestion, and their polyphenol and OSC profiles were determined by ultra-high-performance liquid chromatography coupled with high-resolution mass spectrometry (UHPLC-HRMS). Although to a lesser extent than in the fresh onion, the phenolic compounds in the black variety remained stable during the digestion process, presenting a higher bioaccessibility index (BI) with recovery corresponding to 41.1%, compared with that of fresh onion (23.5%). As for OSCs, apart from being more stable after the digestion process, with a BI of 83%, significantly higher quantities (21 times higher) were found in black onion than in fresh onion, suggesting that the black onion production process has a positive effect on the OSC content. Gallic acid, quercetin, isorhamnetin, and ɣ-glutamyl-S-(1-propenyl)-L-cysteine sulfoxide were the most bioaccessible compounds in fresh onion, while isorhamnetin, quercetin-diglucoside, ɣ-glutamyl-S-methyl-L-cysteine sulfoxide and methionine sulfoxide were found in black onion. These results indicate that OSCs and polyphenols are more bioaccessible in black onion than in fresh onion, indicating a positive effect of the processing treatment.

## 1. Introduction

Onion (*Allium cepa* L.), a member of *Allium* genus, is frequently used in gastronomy around the world and is one of the main ingredients of the Mediterranean diet [1,2]. This product originated in Central Asia, from where its cultivation and consumption spread to the rest of the world [3]. Many medicinal properties have been attributed to members of the *Allium* family throughout history. These properties have made them the focus of many research studies into the relationship between their consumption and the prevention of diseases such as stomach, colorectal, prostate, and breast cancers [4]. Preventing and treating other chronic diseases such as diabetes, cardiovascular diseases, obesity, and metabolic syndromes are some benefits of the regular consumption of *Allium* vegetables, likely due to the presence of biofunctional compounds (polyphenols and OSCs) [5]. Black onion is a novel product that has been developed by the food industry by processing (aging) raw onion under temperature- and humidity-controlled conditions using no artificial additives. The heating treatment (at 60–80 °C) used to produce this product has been shown to affect its phytochemical composition. Up to 12-fold decreases have been found in flavonoid content, while, in contrast, the OSC content is higher [6]. Black onion shows a series of significant compositional and organoleptic modifications in comparison to the original product, such as a sweeter taste, the lack of a spicy sensation (pungency), and a black-brown colour. The main changes with regard to its composition are a lower phenolic compound content and significant increases in isoalliin, fructose, glucose, and tartaric acid [6]. The elimination of the undesirable characteristics of fresh onion at the sensory and digestive levels enhances consumer acceptance of black onions, increasing the probability of their consumption. To date, there have been no studies on the potential benefits of consuming black onion. However, in-vitro studies have shown that black shallot extract presents higher anticancer and anti-inflammatory activities than fresh onion extract when evaluated in cell lines [7]. Indeed, quercetin, the main polyphenol found in black onion, is related to some beneficial effects, such as an antiproliferative effect on ovarian, breast, and colon cancer cells [8] and a protective effect against certain pathologies related to lipid metabolism, such as atherosclerosis and diabetes [9]. Moreover, isoalliin, the major OSC in both fresh and black onion, and other OSCs have been found to present health-promoting benefits [10]. However, to exert beneficial effects in vivo, polyphenols and OSCs from onion must be bioaccessible, be released from the food matrix, and be ready for absorption [11,12] into the gastrointestinal tract. in-vitro digestion models are a good tool for evaluating the bioaccessibility of biofunctional compounds [13,14], including polyphenols and OSCs. For instance, the literature reports the bioaccessibility of polyphenol compounds in different food matrices, including apples [15], blueberries [16], oranges [17], fresh and black garlic [18], and fresh onion [19]. However, to the best of our knowledge, the effect of in-vitro gastrointestinal digestion on the bioaccessibility of the individual profiles of phenolic compounds and OSCs from black onion remains unknown. Therefore, the aim of this study was to evaluate the bioaccessibility of both polyphenols and OSCs from black and fresh onion during simulated gastrointestinal digestion by monitoring them using ultra-high-performance liquid chromatography coupled with high resolution mass spectrometry (UHPLC-HRMS) and to investigate the possible impact of the transformation process during the production of black onion on bioaccessibility. 

## 2. Materials and Methods

### 2.1. Chemicals

α-Amylase enzymes from human saliva (937 units/mg protein), pepsin (500 units/mg protein), pancreatin from porcine pancreas (4 × UPS), bile salts, and calcium chloride were purchased from Sigma-Aldrich (Madrid, Spain). HCl was obtained from Merck (Darmstadt, Germany), and NaOH was acquired from Fisher Scientific (Madrid, Spain). Sodium bicarbonate and ammonium carbonate were purchased from Sigma-Aldrich (Madrid, Spain), sodium chloride and magnesium chloride hexahydrate were purchased from Fisher Scientific (Madrid, Spain), and potassium dihydrogen phosphate was obtained from VWR International Eurolab (Barcelona, Spain). Reference flavonoid compounds including isorhamnetin, luteolin, quercetin, quercetin-3-O-glucoside alliin and s-allyl-L-cysteine (SAC) together with formic acid were acquired from Sigma-Aldrich (Madrid, Spain). Ammonium formate, ammonium acetate, and ethanol were obtained from Sigma-Aldrich. Acetonitrile (LC-MS grade) and methanol (LC-MS grade) were obtained from Panreac (Barcelona, Spain). 

### 2.2. Materials and Sample Preparation

Fresh and black “Shallot” onions (*Allium cepa* var. aggregatum) were obtained from a local supplier (La Abuela Carmen^®^). One batch of fresh onion (5 kg) was divided equally into two groups, one being used to obtain black onion. The black onion samples were obtained by an optimized process that combines heat treatment with controlled humidity conditions, enabling a product with different organoleptic properties to be obtained, as previously described [6]. The fresh and black onions were peeled and ground using liquid nitrogen with cryogenic grinder mill equipment to obtain a final particle size of 10 µm (Freezer Mill model 6870, Fisher Scientific, Waltham, MA, USA) and stored at −80 °C until the simulated gastrointestinal digestion process.

### 2.3. Simulated Gastrointestinal Digestion and Evaluation of Bioaccesibility

An in-vitro oral, gastric, and intestinal digestion model, previously reported by Juániz et al. [20], was adapted to obtain a bolus with the right consistency to perform the in-vitro digestion experiments [21]. Briefly, 2 g of each lyophilized onion sample was weighed in a 100 mL amber glass bottle. The whole process was performed in a stirred water bath (Unitronic Reciprocating Shaking Bath model 6032011, J.P. Selecta, Barcelona, Spain) at 37 °C in triplicate. During the oral phase, simulated salivary fluids (SSFs) (Table 1) were used. A total of 14 mL of SSF solution was added to the samples, together with 250 μL of an α-amylase (300–1500 U/mg protein) solution (1.3 mg/mL), 0.1 mL of 0.3 M CaCl_2_, and 5.65 mL of distilled water. The mixture was shaken at 37 °C for 30 min. For the gastric phase, simulated gastric fluids (SGFs) (Table 1) were used. After the oral phase, it was necessary to adjust the fluids to pH 3 with 1 M HCl solution. Then, 15 mL of SGF solution was added to the samples, together with 1.19 mL of a pepsin (3.2–4.5 U/mg protein) solution, 0.01 mL of 0.3 M CaCl_2_, and 3.8 mL of distilled water. The pepsin solution was prepared with 1 g of pepsin in 10 mL of 0.1 M HCl. The mixture was incubated at 37 °C for 120 min. For the intestinal phase, simulated intestinal fluids (SIFs) (Table 1) were used. After the gastric phase, 22 mL of SIF solution was added to the samples, together with 10 mL of pancreatin (4 × UPS) solution (8 mg/mL), 5 mL of bile salts (25 mg/mL), 0.08 mL of 0.3 M CaCl_2_, and 9.92 mL distilled water. Then, 1 M NaOH solution was used to adjust the solution to pH 7. The mixture was incubated for 120 min at 37 °C. Aliquots of the digested samples were taken before oral digestion (BOD) and after every stage of the digestion process: oral, gastric, and intestinal digestion (AOD, AGD, and AID, respectively). These samples were lyophilized and stored at −80 °C until polyphenol and OSC extraction and analysis. The bioaccessibility indices were calculated as percentages of the initial content (before oral digestion, BOD) of the compound (polyphenol or OSC) after simulated gastrointestinal digestion (before oral digestion, BOD) [22,23]. 

### 2.4. Polyphenol and Organosulfur Compound Extraction and Analysis 

Samples from the in-vitro gastrointestinal digestion were extracted in triplicate following the previously optimized and validated procedure reported by Moreno-Rojas et al. [24]. The polyphenols and OSCs in the fresh and black onion extracts were analysed using an UHPLC-HRMS mass spectrometer system (Thermo Scientific, San José, CA, USA) comprising a UHPLC pump, a PDA detector scanning from 200 to 600 nm, and an autosampler operating at 4 °C (Dionex Ultimate 3000 RS, Thermo Corporation, San José, CA, USA). The chromatographic characteristics of the separation of the polyphenols and OSCs, as well as the details of their identification and quantification, were previously [18,24]. 

### 2.5. Statistical Analysis

The results are expressed as the mean of three replicates measured for each sample. Multiple comparisons were performed using a one-way ANOVA using R statistics software (v. 3.6.3) to identify significant differences between the phases of the simulated gastrointestinal digestion, with significance being established at *p* < 0.05. Next, Fisher’s LSD pairwise comparison was performed on the data. A principal component analysis (PCA) was performed as an unsupervised method using SIMCA software (v.15.0.2) to determine whether the overall changes in the profiles of polyphenols and OS compounds were different enough to distinguish between the simulated gastrointestinal digestion stages and product types (fresh or black onion).

## 3. Results and Discussion

### 3.1. Changes in Polyphenolic Contents of Fresh and Black Onions after Simulated Gastrointestinal Digestion and Bioaccesibility

A total of 17 polyphenols were identified and quantified in the fresh onion samples before simulated gastrointestinal digestion was carried out. Flavonoids were the main ones, accounting for 86.5% of the total content, while phenolic acids represented 13.5%. Overall, quercetin diglucoside (39.7%), quercetin 4-*O*-glucoside (23.6%), vanillic acid (11.6%), and myricetin (9.9%) were the main phenolic compounds in the nondigested fresh onion samples (Table 2). These results are in line with those of Böttcher et al. [25], who reported, among others, quercetin glycosides with glucosyl moieties in 4′-*O* and 3-*O* positions as being the main flavonoids in red and yellow onion cultivars. Regarding black onion, a total of seven polyphenols were identified and quantified, the main type being free quercetin, which accounted for 94% of the total content (Table 2). The remaining polyphenols, representing between 2.6 and 0.06% of the total, were isorhamnetin followed by luteolin, quercetin-diglucoside, quercetin-3-*O*-glucoside, quercetin-4-*O*-glucoside, and isorhamnetin-4-*O*-glucoside (Table 2). This difference in phenolic composition between onion products is mainly attributed to the changes occurring during the black onion elaboration process, as previously reported by Moreno-Ortega et al. [6]. They observed that the heating and humidity conditions used to obtain black onion from fresh onion have an important impact on its physicochemical composition, with free quercetin being the main compound found in black onion from three onion varieties (94% for “Shallot”, 99% for “Chata”, and 99% for “Echalion”). These decreases in the phenolic content during the production of black onion are arguably attributed to the oxidation of flavonoids to semi-quinoid intermediates and the respective quinones, which normally react further with other quinones to produce dark melanin pigments [26] or with proteins to produce dark polymers [27].

The effect of in-vitro gastrointestinal digestion on the concentrations of individual polyphenols in fresh and black onions is shown in (Table 2). A gradual decrease is observed in the total concentration of polyphenols from the buccal phase (AOD) to the intestinal phase (AID) in both onion products. 

For a more in-depth exploration of the stability of polyphenols from fresh and black onions over the different stages of simulated gastrointestinal digestion and to determine the impact of the elaboration process of black onion on the bioaccessibility of polyphenol compounds, a principal component analysis (PCA) was performed (Figure 1). The first PCA (PC1) described 68% of the total variability (Figure 1A) and showed a clear discrimination between the nondigested and digested black and fresh onion samples. This discrimination was attributed to the presence of specific compounds, including vanillic acid, morin, epigallocatechin, myricetin, quercitrin, rutin, and isorhamnetin diglucoside in fresh onion, while compounds such as luteolin and quercetin-3-*O*-glucoside were characteristic of black onion (Figure 1B). PC2 explained 24% of the total variability and highlighted the significant impact of the gastrointestinal digestion process on the polyphenol profiles of both kinds of onion (Figure 1A). Fresh onions seem to be significantly more greatly affected by the digestive process than black ones, with their polyphenol concentration decreasing from 2129 to 500 nmol/g FW, so that 23% of the total polyphenol content remained; meanwhile 41% of the total polyphenol content in the black onion remained almost intact after the digestion process (from 50 to 21 nmol/g FW) (Table 2). Focusing on fresh onion, the oral phase had the greatest impact on its polyphenol content, followed by gastric digestion, with the intestinal digestion phase being the phase with smallest impact on gastrointestinal stability (Table 2, Figure 2). For instance, oral digestion had a very negative effect on the concentration of flavonoids, but not phenols, the former decreasing by 69% from the initial value. The concentrations of specific compounds, such as morin, quercetin diglucoside, rutin, quercetin-4′-*O*-glucoside, quercitrin and isorhamnetin diglucoside, decreased during the oral digestion of fresh onion, probably as a consequence of the hydrolysis of quercetin and isorhamnetin glucosides. Consequently, there were increases in the free quercetin (1.7-fold) and isorhamnetin (2.1-fold) contents. These results should be considered with caution as the timing of our oral phase process (30 min) did not mimic physiological conditions and, therefore, it is impossible to conclude that the oral phase has a great impact on polyphenol stability. However, the results are somewhat suitable for our purpose, which was to compare the bioaccessibility of polyphenols in two food products: fresh and black onion. 

The increases in free quercetin and isorhamnetin continued during gastric digestion, mainly as a result of their partial hydrolysis by the action of pepsin and the pH. A low pH and gastric enzymes could lead to the hydrolysis of proteins and carbohydrates bound to flavonoids, thus improving their extractability and boosting their hydrolysis, which facilitates the release of aglycones from *O*-glycosides during their passage through the stomach [28,29]. In addition, the pronounced decay in flavonoid content, more so in fresh onion than in black onion, is arguably due to the propensity of flavonoids to interact with other matrix components such as carbohydrates or lipids, as indicated by Gonzales et al. [30], or to the propensity of the digestive enzymes to form complexes, as suggested by De Santiago et al. [31] and Su et al. [32]. The phenolic acids in fresh onion were resistant to the oral phase conditions but were greatly affected during the gastric phase, with only 26.5% remaining (Table 2). Likewise, intestinal digestion resulted in a significant increase (almost 3-fold) in the concentration of most of the phenolic acids in fresh onion compared with those obtained after the gastric phase. This can be explained by considering their release from the food matrix after the activity of enzymes such as pancreatin and pH changes in the duodenum [33]. 

The polyphenols in black onion were also affected by the digestion process, as Table 2 shows. As the main polyphenol in black onion is quercetin, the total polyphenol content during gastrointestinal digestion is highly influenced by the stability of this compound. Its concentration showed a gradually decrease during the three steps of gastrointestinal digestion, from 47 nmol/g FW before oral digestion to 18 nmol/g FW after the intestinal digestion phase, with 39.3% of its initial content remaining. Overall, the different effects of in-vitro digestion on the polyphenol profiles of fresh and black onion highlight the importance of the food matrix and its interaction with other compounds, particularly in terms of how polyphenols are released during the digestion process, as previously reported by other authors [34,35,36]. The most bioaccessible compounds in fresh onion were found to be gallic acid (316.7% bioaccessibility index (BI)), quercetin (165% BI), and isorhamnetin (210.7% BI), while in black onion, quercetin-*O*-glucoside (95.4% BI) and isorhamnetin (81.6% BI) were more bioaccessible after in-vitro gastrointestinal digestion (Table 2). These results indicate that the compounds remaining after intestinal digestion—significant quantities of vanillic acid and quercetin-4-*O*-glucoside in fresh onion and free quercetin in black onion—potentially cross the small intestine and reach the colon, where they are subjected to microbiota-mediated metabolism prior to absorption.

### 3.2. Changes in Organosulfur Compound Profiles of Fresh and Black Onion after Simulated Gastrointestinal Digestion

A total of 24 OSCs were identified and quantified in the fresh onion samples before oral digestion (Table 3), the predominant ones being γ-glutamyl-S-(2-propenyl) cysteine sulfoxide (G2PCS) (2615 nmol/g FW), isoalliin (1348 nmol/g FW), γ-glutamyl-S-(2-carboxypropyl) cysteine–glycine (884 nmol/g FW), γ-glutamyl-S-allyl-L-cysteine (GSAC) (799 nmol/g FW), and propanethial sulfoxide (596 nmol/g FW), accounting for 74.0% of the total initial content (Table 3). The remaining OSCs are listed in Table 3. Variations were found in the OSC profile of black onion samples compared with that of fresh onion, the main differences being found for isoalliin (53,117 nmol/g FW), propanethial sulfoxide (10,663 nmol/g FW), methionine sulfoxide (1073 nmol/g FW), and G2PCS (861 nmol/g FW), representing 98.9% of the total OSC content. The remaining OSCs were γ-glutamyl-S-methyl cysteine sulfoxide (GSMCS), γ-glutamyl-S-propyl cysteine sulfoxide (GSPC), S-(S-propyl) cysteine, methiin, and propiin. These results are in line with those previously reported by Moreno-Rojas et al. [24], who showed that G2PCS, isoalliin, and γ-glutamyl-S-(2-carboxypropyl) cysteine–glycine are the main OSCs in fresh shallot, while isoalliin and G2PCS are the main ones in black onion. 

Table 3 shows the impact of in-vitro gastrointestinal digestion on the stability of OSCs in fresh and black onion. To better understand the impacts of the different digestive phases on the OSC concentration of fresh and black onion, a Principal Component Analysis (PCA) was performed (Figure 3). PC1, which explained 76.4% of the total variability, showed a clear discrimination between fresh and black onion samples (Figure 3A). This differentiation was ascribed principally to the presence of S-alk(en)yl-L-cysteine (SAC) derivatives, including methiin, propiin, isoalliin, S-(S-propyl) cysteine, propanethial S-oxide, and methionine sulfoxide, as well as γ-glutamyl-S-propyl cysteine sulfoxide (GSPC), in the black onion samples (Table 3, Figure 4). Meanwhile fresh onion was characterized by the presence of other specific γ-glutamyl-S-alk(en)yl-L-cysteine derivatives (GSAks) and SAC derivatives including γ-glutamyl-S-(2-carboxypropyl) cysteine-glycine, γ-glutamyl-S-allyl-L-cysteine (GSAC), γ-glutamyl-S-(2-propenyl) cysteine sulfoxide (G2PCS), γ-glutamyl-S-(propyl) cysteine sulfoxide, S-(2-carboxypropyl) cysteine-glycine, S-propyl-L-cysteine, S-allyl-L-cysteine (SAC), alliin, and isoalliin (Figure 3B). According to Moreno-Ortega et al. [6], during the thermal processing of black onion, many GSAk derivatives are transformed into simple and intermediate volatile compounds, thus decreasing their concentrations. It is worth noting that the total OSC content was significantly higher in black onion (66,452 nmol/g FW) than in fresh onion (8432 nmol/g FW) before the digestive process, mainly due to the high concentration of isoalliin, which totalled 1348 nmol/g FW in fresh onion and 53,117 nmol/g FW in black onion. PC2 explained 14.1% of the total variance and provided clear discrimination between samples belonging to the different gastrointestinal digestion phases of both kinds of onion (Figure 3A). In particular, the OSCs in fresh onion were more strongly affected by the oral phase, with the SAC derivatives (with 60.1% remaining, ranging from 45.8% for S-propyl-L-cysteine (deoxypropiin) to 85.9% for S-allyl-L-cysteine, SAC) being more greatly affected than the GSAk derivatives (with 70.2% remaining, ranging from 42.5% for γ-glutamyl-S-(propyl) cysteine sulfoxide to 95.0% for γ glutamyl-S-(propyl) cysteine, GSPC) (Figure 4). However, there were significant decreases in the recovery of compounds such as methiin (9.2%), propiin (4.6%), and alliin (not quantified), since they are the main substrates of the alliinase enzyme, which is found in the composition of the members of the *Allium* genus, as reported by Keusgen et al. [37]. The decrease in the total OSC content continued during gastric and intestinal digestion, with mean recoveries of 49.2 and 30.8%, respectively (Table 3). Meanwhile, the intestinal phase had the greatest impact on the OSC content in black onion. This was not due to the decrease in the total OSC content but mainly to the significant increases in specific OSCs such as methionine sulfoxide (168%) and GSMCS (160.3%). The decreases in methiin, propiin, and S-(S-propyl) cysteine during this stage were mainly due to the fact that these smaller structures are very unstable under the alkaline conditions present during intestinal digestion [38].

The methoxidation reaction of methionine during simulated gastrointestinal digestion could explain the significant increase in methionine sulfoxide [39], while the increase in the GSMCS level could be due to the oxidation of its precursor, ɣ-glutamyl-S-methyl-L-cysteine (GSMC), during the in-vitro process, a factor that has been identified previously in black onion [6,24]. 

Overall, fresh onion was significantly more greatly affected by the digestive process than black onion (Figure 4), with its OSC content decreasing by 69%, from 8432 to 2594 nmol/g FW. In black onion, the decrease in OSCs from 66,452 to 55,153 nmol/g represented a total loss of 17% (Table 3). The stability of the main OSCs during the stages of digestion could be explained by the inactivation of the alliinase enzyme during the production process of black onion, in which temperatures above 60 °C are reached, preventing its interaction with these compounds, as reported by Méndez et al. [40]. Moreover, the fresh onion was identified as γ-glutamyl-S-(1-propenyl)-L-cysteine sulfoxide (G1PCS) (206.3%), S-(S-propyl) cysteine (66.7%), and methionine sulfoxide (65.5%), with G2PCS, γ-glutamyl-S-(2-carboxypropyl) cysteine–glycine, GSAC, isoalliin, propanethial sulfoxide, deoxypropiin, and S (2 carboxypropyl) cysteine–glycine being the main OSCs remaining after in-vitro digestion. GSMCS and methionine sulfoxide were identified as the most bioaccessible OSCs in black onion, although isoalliin was predominant, accounting for 80.1% of the total content. These results suggest that these OSCs, as occurs with the polyphenols, will potentially cross the small intestine and reach the colon, where they will undergo microbiota-mediated metabolism prior to absorption.

## 4. Conclusions

This study evaluated the effects of simulated gastrointestinal digestion on the bioaccessibility of polyphenols and OSCs in black onion compared with its fresh counterpart. During the digestive process, there was a decrease in the concentration of glycosylated flavonoids in fresh onion but a significant increase in the contents of free quercetin and isorhamnetin, the bioavailable forms of these compounds at the colonic level. These polyphenols showed the highest bioaccessibility indexes in fresh onion (165 and 210.7%, respectively) along with gallic acid (316.7%), vanillic acid (70.4%), and ferulic acid (62.5%). Meanwhile in black onion, the lower initial polyphenol content compared with that of fresh onion progressively decreased during in-vitro digestion, showing a total bioaccessibility index of 41.1%. The OSC content of the fresh onion was affected to a greater extent during the oral and intestinal stages than in the gastric stage, mainly because the alliinase enzyme was more active at the neutral pH found during the oral and intestinal stages. However, during the digestion of black onion, different behaviours were observed among the three stages, with a more stable trend being found for the OSC concentrations and a total bioaccessibility index of 83.3%. This greater stability could be explained by the fact that the temperatures used to produce black onion may inactivate the alliinase enzyme. Therefore, it seems that the black onion production process has a positive effect on the bioaccessibility of OSCs, with propanethial sulfoxide, isoalliin, GSMCS, and methionine sulfoxide being the OSCs that were most readily absorbed and transformed in the large intestine. 

## Figures and Tables

**Figure 1 foods-10-00337-f001:**
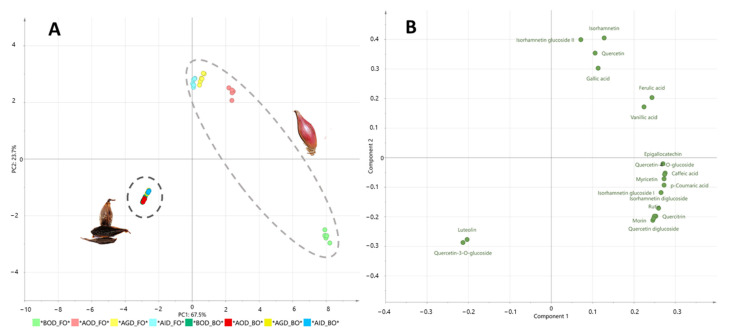
Scores (**A**) and loadings (**B**) obtained in the Principal Component Analysis (PCA) comparing data from polyphenols in fresh and black onion during simulated gastrointestinal digestion.

**Figure 2 foods-10-00337-f002:**
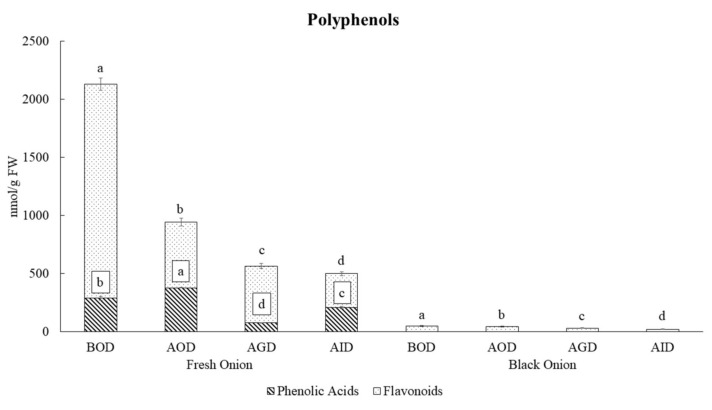
Quantities of phenolic acids and flavonoids during in-vitro simulated gastrointestinal digestion of fresh and black onion. Data are expressed in nmol/g FW as mean values (*n =* 3). Different letters (one-way ANOVA) denote statistically significant differences between the stages of simulated gastrointestinal digestion (*p*-value < 0.05).

**Figure 3 foods-10-00337-f003:**
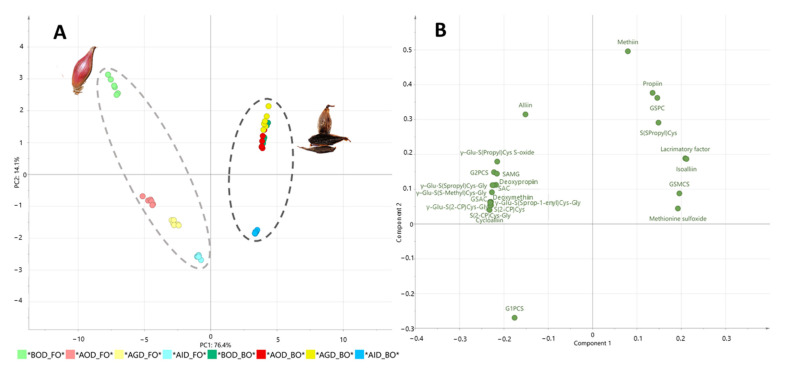
Scores (**A**) and loadings (**B**) of the PCA comparing data from organosulfur compounds of fresh and black onion during simulated gastrointestinal digestion.

**Figure 4 foods-10-00337-f004:**
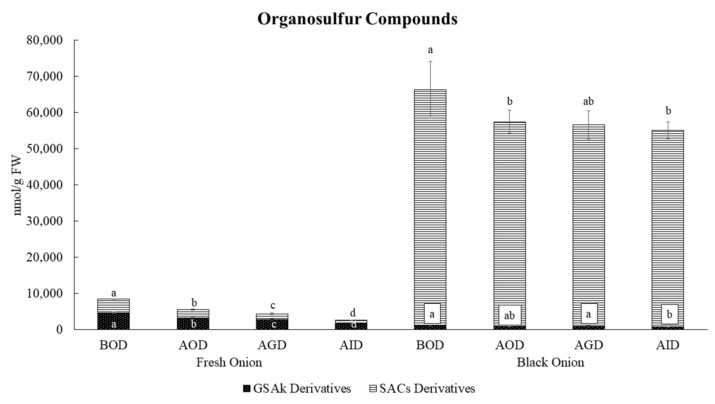
Quantities of organosulfur compounds during in-vitro simulated gastrointestinal digestion of fresh and black onion. Data are expressed in nmol/g FW as mean values (*n =* 3). Different letters (one-way ANOVA) denote statistically significant differences between the stages of simulated gastrointestinal digestion (*p*-value < 0.05).

**Table 1 foods-10-00337-t001:** Details of the fluids used in the simulated gastrointestinal digestion.

Solution	Concentration	SSFs	SGFs	SIFs
	Molarity	mL	mL	mL
MagCl_2_(H_2_O)_6_	0.15	0.313	0.250	0.688
KCl	0.50	9.438	4.313	4.250
KH_2_PO_4_	0.50	2.313	0.563	0.500
(NH_4_)_2_CO_3_	0.50	0.038	0.313	0.688
NaHCO_3_	1.00	4.250	7.813	26.563
NaCl	2.00	-	7.375	6.000
Distilled Water	-	233.650	229.375	211.313
Final Volume		250	250	250

SSFs: simulated salivary fluids; SGFs: simulated gastric fluids; SIFs: simulated intestinal fluids.

**Table 2 foods-10-00337-t002:** Concentration (nmol/g FW) of polyphenols found in fresh and black onion samples at different stages of simulated gastrointestinal digestion. Data are expressed as mean values (*n* = 3).

Compounds	BOD	AOD	% Recovery	AGD	% Recovery	AID	% Recovery-Bioaccessibility	*p*-Value
	*Fresh Onion*							
Phenolic Acids								
p-Coumaric acid	2.48 a	0.93 b	37.5	0.36 c	14.5	0.32 c	12.9	***
Vanillic acid	274 b	365 a	133.2	70 d	25.5	193 c	70.4	***
Gallic acid	3.6 c	4.9 b	136.1	2.6 d	72.2	11.4 a	316.7	***
Caffeic acid	2.05 a	1.01 b	49.3	0.43 c	21.0	0.27 d	13.2	***
Ferulic acid	4.8 b	5.3 a	110.4	2.8 c	58.3	3.0 c	62.5	***
Total Phenolic Acids	287 b	377 a	131.4	76 d	26.5	208 c	72.5	***
Flavonoids								
Morin	56.46 a	2.99 b	5.3	1.69 b	3.0	0.97 b	1.7	***
Quercetin	80 c	81 c	101.3	184 a	230.0	132 b	165.0	***
Epigallocatechin	1.56 a	0.54 c	34.6	0.71 b	45.5	0.20 d	12.8	***
Isorhamnetin	28 c	60 b	214.3	81 a	289.3	59 b	210.7	***
Myricetin	211.4 a	86.1 b	40.7	58.1 c	27.5	11.0 d	5.2	***
Quercitrin	9.1 a	1.3 b	14.7	0.171 c	1.9	0.077 c	0.8	***
Quercetin-4-*O*-glucoside	503 a	247 b	49.1	106 c	21.1	55 d	10.9	***
Isorhamnetin glucoside I	4.06 a	1.81 b	44.6	0.27 c	6.7	0.13 c	3.2	***
Isorhamnetin glucoside II	nd	15.52 a	-	13.09 b	-	4.84 c	-	***
Rutin	12.11 a	0.96 b	7.9	0.53 b	4.4	0.58 b	4.8	***
Quercetin diglucoside	846 a	52 b	6.1	36 b	4.3	23 b	2.7	***
Isorhamnetin diglucoside	90.6 a	16.8 b	18.5	6.5 c	7.2	4.1 c	4.5	***
Total Flavonoids	1842 a	567 b	30.8	488 c	26.5	292 d	15.9	***
Total	2129 a	944 b	44.3	564 c	26.5	500 d	23.5	***
	*Black Onion*							
Quercetin	47 a	39 ab	83.3	28 bc	59.6	18c	39.3	***
Isorhamnetin	1.32 a	0.89 b	67.9	0.85 b	64.7	1.08 ab	81.6	**
Luteolin	0.23 a	0.22 a	98.4	0.12 b	51.3	0.14 b	61.1	**
Quercetin diglucoside	0.20 ab	0.21 a	106.5	0.15 b	76.7	0.19 ab	95.4	*
Quercetin-3-*O*-glucoside	0.63 a	0.60 a	94.6	0.57 a	90.6	0.42 b	66.6	**
Quercetin-4-*O*-glucoside	0.81 a	0.57 b	70.2	0.81 a	100.5	0.35 c	43.7	***
Isorhamnetin-4′-*O*-glucoside	0.031 a	0.031 a	98.8	nq	-	nq	-	***
Total	50 a	42 b	83.0	30 c	60.7	21 d	41.1	***

Different letters (one-way ANOVA) denote significant differences (*p <* 0.05) between the four stages for the same compound. Ns: non-significant; * *p*-value < 0.05; ** *p*-value < 0.01; *** *p*-value < 0.001. nq: not quantified; nd: not detected. BOD: before oral digestion; AOD: after oral digestion; AGD: after gastric digestion; AID: after intestinal digestion.

**Table 3 foods-10-00337-t003:** Concentrations (nmol/g FW) of organosulfur compounds in fresh and black onion samples at different stages of simulated gastrointestinal digestion. Data are expressed as mean values (*n =* 3).

Compounds	BOD	AOD	% Recovery	AGD	% Recovery	AID	% Recovery-Bioaccessibility	*p*-Value
	*Fresh Onion*							
ɣ-Glutamyl-S-alk(en)yl-L-cysteine derivatives (GSAk)								
γ–Glutamyl-S-(2-carboxypropyl) cysteine–glycine	884 a	620 b	70.2	499 c	56.4	253 d	28.6	***
γ–Glutamyl-S-(S-1-propenyl) cysteine–glycine	3.09 a	2.07 b	67.0	1.69 c	54.7	0.94 d	30.5	***
γ–Glutamyl-S-(S-methyl) cysteine–glycine	5.77 a	4.38 b	76.0	2.39 c	41.5	1.35 d	23.4	***
γ–Glutamyl-S-(S-propyl) cysteine–glycine	17.7 a	12.9 b	73.0	7.8 c	43.9	2.8 d	15.9	***
γ-Glutamyl-S-allyl-L-cysteine (GSAC)	799 a	574 b	71.9	437 c	54.7	225 d	28.1	***
γ–Glutamyl-S-(propyl) cysteine (GSPC)	13.78 a	13.09 a	95.0	0.24 b	1.8	nd	0.0	***
γ-Glutamyl-S-methyl cysteine sulfoxide (GSMCS)	15.0 a	9.3 b	62.2	9.1 b	60.8	7.0 c	46.4	***
γ- Glutamyl-S-(2-propenyl) cysteine sulfoxide (G2PCS)	2615 a	1908 b	72.9	1566 c	59.9	1102 d	42.1	***
γ–Glutamyl-S-(propyl) cysteine sulfoxide	323 a	137 b	42.5	94 c	29.1	53 d	16.5	***
γ−Glutamyl-S-(1-propenyl)-L-cysteine sulfoxide (G1PCS)	2.1 c	4.6 a	213.7	2.9 b	136.0	4.4 a	206.3	***
Total GSAk derivatives	4679 a	3286b	70.2	2620 c	56.0	1649 d	35.3	***
S-Alk(en)-yl-L-cysteine derivatives (SACs)								
S-(2-carboxypropyl) cysteine-glycine	333 a	242 b	72.5	198 c	59.3	97 d	29.1	***
S-methyl-cysteine (deoxymethiin)	41.4 a	33.9 b	81.9	20.7 c	50.1	11.1 d	26.8	***
S-Propyl-L-cysteine (deoxypropiin)	359 a	165 b	45.8	128 c	35.6	121 c	33.7	***
S-Allyl-L-cysteine (SAC)	343 a	295 b	85.9	90 c	26.3	68 c	19.9	***
S-allylmercaptoglutathione	0.31 a	0.19 b	60.3	0.17 b	55.6	nd	0.0	***
S-(S-propyl) cysteine	20.9 a	14.3 b	68.3	8.0 c	38.5	13.9 b	66.7	***
S-(2-carboxypropyl) cysteine	11.2 a	9.2 b	82.2	5.6 c	50.4	3.3 d	29.6	***
Alliin	330.7 a	nq	-	nq	-	nd	0.0	***
Isoalliin	1348 a	943 b	70.0	661 c	49.0	379 d	28.1	***
Propanethial sulfoxide (lacrimatory factor)	596a	418 b	70.1	296 c	49.6	175 d	29.3	***
Methyl-L-cysteine sulfoxide (methiin)	160.2 a	14.7 b	9.2	15.8 b	9.9	6.3c	3.9	***
S-propyl-cysteine sulfoxide (propiin)	31.5 a	1.5 b	4.6	0.6b c	1.9	nd	0.0	***
Cycloalliin	166 a	114 b	68.7	93 c	55.8	62 d	37.6	***
Methionine sulfoxide	12.2 a	6.7 c	54.6	6.3 c	51.2	8.0 b	65.5	***
Total SACs derivatives	3755 a	2256 b	60.1	1523 c	40.6	945 d	25.2	***
Total	8433 a	5543 b	65.8	4142 c	49.2	2594 d	30.8	***
	*Black Onion*							
ɣ-Glutamyl-S-alk(en)yl-L-cysteine derivatives (GSAk)								
γ-Glutamyl-S-methyl cysteine sulfoxide (GSMCS)	115 c	104 c	90.4	138 b	119.7	185 a	160.3	***
γ-Glutamyl-S-(2-propenyl) cysteine sulfoxide (G2PCS)	861 a	753 a	87.4	800 a	92.9	603 b	70.0	***
γ–Glutamyl-S-propyl cysteine sulfoxide (GSPCS)	62 b	63 b	101.8	90 a	144.6	nd	0.0	***
Total GSAk derivatives	1039 a	921 ab	88.6	1028 a	99.0	788 b	75.9	*****
S-Alk(en)-yl-L-cysteine derivatives (SACs)								
S-(S-propyl) cysteine	178 a	175 a	98.5	90 b	77.9	nd	0.0	***
Isoalliin	53,117 a	45,859 ab	86.3	47,859 ab	90.1	44,199 b	83.2	*
Propanethial sulfoxide (Lacrimatory factor)	10,663 a	9299 ab	87.2	8176 b	76.7	8356 b	78.4	**
S-methyl-cysteine sulfoxide (methiin)	181 a	179 a	99.0	208 a	114.8	nd	0.0	***
S-propyl-cysteine sulfoxide (propiin)	83 b	77 c	92.6	151 a	181.4	nd	0.0	***
Methionine sulfoxide	1073 b	923 b	86.0	1006 b	93.8	1809 a	168.5	***
Total SACs derivatives	65,296 a	56,511 b	86.5	57,489 ab	88.0	54,364 b	83.3	*
Total	66,334 a	57,432 b	86.6	58,517 ab	88.1	55,153 b	83.0	*

Different letters (one-way ANOVA) denote significant differences (*p <* 0.05) between the four stages for the same compound. Ns: non-significant; * *p*-value < 0.05; ** *p*-value < 0.01; *** *p*-value < 0.001. nq: not quantified; nd: not detected. BOD: before oral digestion; AOD: after oral digestion; AGD: after gastric digestion; AID: after intestinal digestion.

## Data Availability

Not applicable.

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
