# Peer review of "Changes in the Organosulfur and Polyphenol Compound Profiles of Black and Fresh Onion during Simulated Gastrointestinal Digestion"

_foods, 2021, doi:10.3390/foods10020337_

Round 1
Reviewer 1 Report
The article is very much like a previous article by the same authors on black garlic (here black onion) and in that sense lacks originality.
Overall the experimental work seems to be conducted seriously, but the interpretation and conclusions are sometimes unexpected, and more importantly there is little consideration of what they mean from a physiological perspective. For example, the abstract states that “phenolic compounds in black onion remained more stable under the digestion process (…) compared to (…) fresh onions” (suggesting that the process at the origin of black onions is beneficial) but without mentioning that the total phenolic compounds concentration at the end of the digestion process is almost 25 times lower for black onion compared to fresh onions.
To be acceptable, the limitations of the model (in particular the oral phase which is nothing like the real situation) should be clearly indicated, and the physiological relevance of findings should be discussed.
The English language should be reviewed.
Below are other comments.
Line 22-23: Compounds (…) were more bioaccessible in fresh onions. The use of the word “more” suggests a comparison (with black onion) which is not the case. Rephrase.
Line 58: the word bioavailable should rather be bioaccessible here (at least if the following “be released from the food matrix and be ready for absorption” is a definition).
Line 84: Fresh and black ‘Shallot’ onions (and is lacking)
Line 95-98: this oral phase (2 g of product for 20 ml of SSF, incubation for 30 minutes) is incredibly different from the physiological situation (which would be rather 2g of product for a maximum of 2ml of SSF, incubation for a maximum of 1 minute). When the results are discussed later, it is essential to highlight this difference.
Lines 98-104: activities of the digestive enzymes need to be reported.
Line 110-111: The definition does not explain how bioaccessibility is calculated.
Line 122: It is unclear what this “6 replicates” refers to. Above line 95, it is said that the digestion process is conducted in triplicates. Then, are there 2 analytical replicates for each of the experimental replicates? In that case the statistics should not be performed considering that n=6. Please clarify and adjust the statistical treatment if necessary.
Lien 174-176: here it is absolutely necessary to stress the discrepancy between the in vitro oral phase and the in vivo situation, in order to avoid conveying the wrong message: based on this model, it is impossible to say that the oral phase in real consumption conditions would have a great impact on polyphenol content.
Line 181: citation of this article is wrong: Kahle et al. show that clarified saliva (containing salivary enzymes but not microorganisms) has no impact on glucosides hydrolysis, and conclude that this hydrolysis can be attributed to the oral flora, not to salivary amylase.
Line 183-185: I really cannot see how pepsin can have an impact on the reaction leading to the increase in free quercetin /isorhamnetin. Justify or correct.
Line 214-216: Not only those compounds (in large quantities) but all those present at the end of the GI digestion process can potentially reach the colon. Rephrase.
Line 218: Go to new line before 3.2
Lines 223 and 227: in my view the “represented 26%” and “accounting for 1.1%” are unnecessary… quite obvious with the quoted 74% and 98.9%...
Table 3:
add abbreviation GSPC to top half of the table.
I believe a 0% recovery is wrong when a compound is not quantified (ex: alliin AOD), it should be a missing value.
In contrast, when a compound is “not detected”, then a 0% recovery proxy is appropriate: it is done in the table for example for propiin in fresh onions but not for several compounds in black onions: GSPC, S-(S-propyl) cysteine, methiin, propiin. Correct the table for consistency.
Line 264: same remark, the 0% for alliin is not appropriate.
Line 283: Should it be a full stop between “a 69 % decrease” and “in the black onion”?
Line 285-288: There is still a 0% final recovery for methiin and propiin, explain how this can happen despite alliinase inactivation.
Line 314: I do not understand why isoalliin, GSMCS and methionine sulfoxide are specifically highlighted. It is neither on the basis of final recovery percentage (lacrimatory factor has recovery quite near that of isoalliin), nor on the basis of concentration at the end of the digestive process (lacrimatory factor has a high final concentration). Explain.
Reviewer 2 Report
The manuscript is well written and provides an interesting perspective on the bioaccessibility of selected nutraceutical compounds in onion under different storage conditions; in my opinion, it might constitute a nice contribution to the Journal.
Some minor comments to improve the manuscript are embedded in the text (see PDF)

Round 2
Reviewer 1 Report
It can't be true that the manuscript has been entirely revised by a native English speaker. For example, see this sentence in the abstract:
"Moreover, regarding OSCs, apart from being more stables after the digestion process of black onion, with BI of 83%, there were also in significantly higher quantities (21 times 22 higher) in black onion compared with fresh onion, suggesting a positive effect of the black onion elaboration process with respect to OSCs content".
There are many other examples, especially in newly added sentences.
Line 27. Compounds such as gallic acid, quercetin, isorhamnetin and ɣ-glutamyl-S-(1-propenyl)-L-cysteine sulfoxide were the most bioaccessible events in fresh onion. Events??
Lines 107, 111. I don’t understand why ranges of enzyme activity are reported. It means that enzymes were not characterized precisely for each experiment?? Or are these numbers activities in different batches? Then it means that the amylase activity may vary 5-fold from one experiment to another??
Line 198: citation of this article is in my view still not accurate… The authors’ reply to this comment is: “Kahle et al. report on the effect of oral microbiota during oral digestion, establishing that the hydrolysis of glycosylated polyphenols is greatly dependent on it. Even so, they show that a minor part of this hydrolysis is due to the action of salivary amylase.” To me, Kahke e al. do not report this. Here is (copied and pasted) their exact words on the impact of saliva: “In test series A, the supernatant obtained after centrifuging saliva was used as inoculum. Under these conditions, all the polyphenols investigated were stable and recoveries ranged between 94.4 and 100.3%, indicating that the activity of soluble salivary enzymes was very low.” I’m not sure that a maximum of 5% loss can really translate activity…
